# Simulating Discharge in a Non-Dammed River of Southeastern South America Using SWAT Model

Thais Fujita [1,2,*], Marcos Vinicius Bueno de Morais [1,3], Vanessa Cristina Dos Santos [4], Anderson Paulo Rudke [2,5], Marilia Moreira de Eiras [2], Ana Carolina Freitas Xavier [2,6], Sameh Adib Abou Rafee [1,2,7,8], Eliane Barbosa Santos [9], Leila Droprinchinski Martins [2], Cintia Bertacchi Uvo [7,10], Rodrigo Augusto Ferreira de Souza [11], Edmilson Dias de Freitas [1] and Jorge Alberto Martins [2]

1 Department of Atmospheric Sciences, University of São Paulo, São Paulo 05508-090, Brazil; marcosmorais@utfpr.edu.br (M.V.B.d.M.); samehabou@gmail.com (S.A.A.R.); edmilson.freitas@iag.usp.br (E.D.d.F.)
2 Graduate Program in Environmental Engineering, Federal University of Technology—Paraná, Londrina 86036-370, Brazil; rudke@ufmg.br (A.P.R.); eiras.marilia@gmail.com (M.M.d.E.); anacarolinaf.xavier@gmail.com (A.C.F.X.); leiladromartins@gmail.com (L.D.M.); jmartinseae@gmail.com (J.A.M.)
3 Departamento de Obras Civiles, Universidad Católica de Maule, Talca 3480112, Chile
4 Laboratoire Ecologie Fonctionnelle et Environnement (UMR 5245 CNRS—UT3—INPT), 31326 Castanet Tolosan, France; van.c.dossantos@gmail.com
5 Graduate Program in Sanitation, Environment and Water Resources, Federal University of Minas Gerais, Belo Horizonte 31270-901, Brazil
6 Sustainable Development, Vale Institute of Technology, Belém 66055-090, Brazil
7 Division of Water Resources Engineering, Lund University, Box 118, SE-221 00 Lund, Sweden; cintia.uvo@syke.fi
8 Institute of Natural Resources, Federal University of Itajuba, Itajuba 37500-903, Brazil
9 Meteorology Laboratory, State University of North Fluminense, Macaé 27925-535, Brazil; elianbs@gmail.com
10 Finnish Environment Institute, P.O. Box 140, FI-00790 Helsinki, Finland
11 Superior School of Technology, State University of Amazonas, Manaus 69065-020, Brazil; souzaraf@gmail.com
* Correspondence: fujita.thais@gmail.com

**Abstract:** Within a single region, it is possible to identify opposite changes in flow production. This proved to be the case for several basins in southeastern South America. It remains challenging to the causes this behavior and whether changes in streamflow will continue at current levels or decline in the coming decades. In this study, we used the Soil Water Assessment Tool to simulate monthly river discharge in the Ivaí River Basin, an unregulated medium-sized catchment and tributary of the Upper Paraná River Basin. After calibration, the simulated flow regime for the five streamflow stations based on the Nash-Sutcliffe Efficiency index (NSE) rated four of the streamflow stations Very Good (NSE between 0.86 and 0.89) and only one in the Good index (0.70). The overall flow behavior was well represented, although an underestimation was identified in four monitoring stations. Through assessment of its functionality and limitations in terms of specific flow duration curves percentages, the calibrated model could provide (to managers) the reliability needed for a realistic intervention. The results of this study may assist managers and support public policies for the use of water resources at the Ivaí River basin.

**Keywords:** hydrological modeling; SWAT-CUP; flow duration curve; Ivaí River Basin

## 1. Introduction

Brazil is among the countries with the highest volume of freshwater in the world, and the available water is estimated to be around 78,600 m³/s [1]. Despite its availability, water resources are distributed unequally across the country, as it contains different rainfall regimes, geographical patterns, and distinct biodiversity. Therefore, single governance is

not effective when based on administrative and political boundaries. In this sense, the river basin, as a unit of water resources management, was adopted in 1997 after the promulgation of the National Water Law (n° 9.344/97). Despite that, the National Water Resources Policy (NWRP) was approved in its final form only in 2006.

Although it was only recently approved, the NWRP considers national geographical, social, and economic diversity through region and user sector, employing decentralized planning. Still, it did not reach the entire Brazilian territory. Despite substantial plans and numerous committees, the political and social maturity required to utilize water for multiple purposes has not yet been met. As for governance and political management, their decisions depend on statistical methods found in databases that are difficult to handle or access. This issue was exacerbated by a serious water crisis that prevailed during 2014–2015. That resulted in an exceptionally dry period, with no rainfall during several consecutive weeks and temperatures consistently above average [2]. The water scarcity affected water supply systems in the northern portion of the Upper Paraná River Basin (UPRB), where the larger metropolitan areas, such as São Paulo and Rio de Janeiro, are placed. These areas have higher water demands [3]. The affected UPRB region is also home to several hydropower plants with water storage reservoirs. These are primarily maintained and operated using energy generation optimization, which prioritizes the management of upstream systems. Additionally, due to the depletion of reservoirs, the country activated all available thermal power plants to supply the national electric system and avoid worsening the energy shutdown. The water crisis created strong disputes between industrial hubs and populations regarding water use, forcing extreme measures such as the pumping of dead storage reservoirs reserved for emergency use.

Whether the 2014–2015 crisis was a result of natural climate variability or climate change remains an open question. Regardless, the increase in water consumption due to population growth and economic activities put the water supply at serious risk. This was further accompanied by impairment to ecosystems and biodiversity [4–6]. The need to balance different demands for water resources has grown increasingly evident as more components are incorporated into the concept of water security [7]. In this sense, decentralized assistance to the user sectors compatible with the knowledge of the streamflow regime and physical characteristics of hydrographic basins has become fundamental in managing territories and their respective natural resources, as they also fulfill their socioeconomic and environmental functions. Hydrological models are the best tools available to support decisions by managers under the pressure of crises such as that of 2014–2015, which affected most of the UPRB. However, at the time of that crisis, this powerful tool was not available to aid in decision making.

Although hundreds of papers have shown the successful use of hydrological models in basin management in several parts of the globe, this was not the case for the UPRB. To date, in the well-established hydrological model Soil Water Assessment Tool (SWAT) Literature Database (https://www.card.iastate.edu/swat_articles/ (accessed on 25 November 2021)), we found 149 peer-reviewed studies pertaining to Brazilian territories, of which 53 were located within the UPRB. Of these studies, only 18 were implemented as of the end of 2013, with the largest modeled basin comprising 1710 km$^2$ [8]. Most of the studies were written in the Portuguese language. Of the six studies published in international journals among those cited, it was noted that the largest covered no more than 234 km$^2$ [9]. For instance, the complex Cantareira System alone, which contributes nearly half of the water consumed by the metropolitan area of São Paulo, comprises more than 15,300 km$^2$. Nevertheless, the entire southeastern part of the country suffered from the events of 2014–2015, which affected more than 900,000 km$^2$.

The primary lesson of the 2014–2015 water crisis was the need to prepare tools for the managers of the various UPRB subbasins, including those unaffected by the crisis. Since then, many studies have addressed the variability of the streamflow over the basin [2,3,10–13]. Additionally, several hydrological models of the UPRB were presented along with results discussed based on the main sub-basins [14–16]. Despite progress in the

application of modeling in large-scale basins, its use has not yet been incorporated into the articulation that integrates the main user sectors and resource management bodies. As evidence of this, it was noted that hydrological modeling is not among the auxiliary tools for anticipatory decisions in the face of the most recent drought. Since 2019, an exceptionally long meteorological drought exposed drought-sensitive sectors. It has continued to impact several socioeconomic sectors through considerable decreases in river flows, severely affecting ecosystems [17].

Therefore, the main goal of this study was to set up and calibrate a hydrological model for monthly discharge in a subbasin of the UPRB so that it would be readily available to users and managerial sectors. Proposing a deterministic model and communicating it in detail favors possible solutions to problems and limitations that could then be addressed in international cooperation efforts. Moreover, these solutions could be applied locally—a situation that has not always been within reach of managers, but that is common in the scientific community. By proposing the verification of the calibrated hydrological models' operational adequacy using flow duration curves percentages, greater transparency can be created in the disclosure of the outputs obtained by the modeling regardless of its application. Additionally, by improving interpretations and physical knowledge of the territory, it will be possible to create new information collection methods and databases to direct managers and stakeholders—supporting the exchange of information and acting as a facilitator between various sectors of society. Using this approach to interpret the control points in the main course of the river can allow for the adoption of a basic standard tool for local water resource planning. Such detail could meet the needs of regional managers to apply measures respecting the heterogeneity of smaller basins within the UPRB. This approach could also be reproduced for other watersheds and incorporated into plans and committees to meet political and social maturity expectations.

## 2. Materials and Methods

### 2.1. Study Area

The Ivaí River Basin (IRB) is in southern Brazil and covers an area of 36,589 km² (Figure 1), with a 680 km length. The source is 800 m above sea level. The river flows inland in a northwest direction, assumes a western orientation, and empties into the Paraná River, featuring one of the major tributaries of its left margin [18]. The elevation measurement ranges from 187 to 1335 m. Ivaí basin also belongs to the Itaipu Incremental Basin that regulates Itaipu Dam, the second-largest hydropower plant in the world in terms of installed capacity.

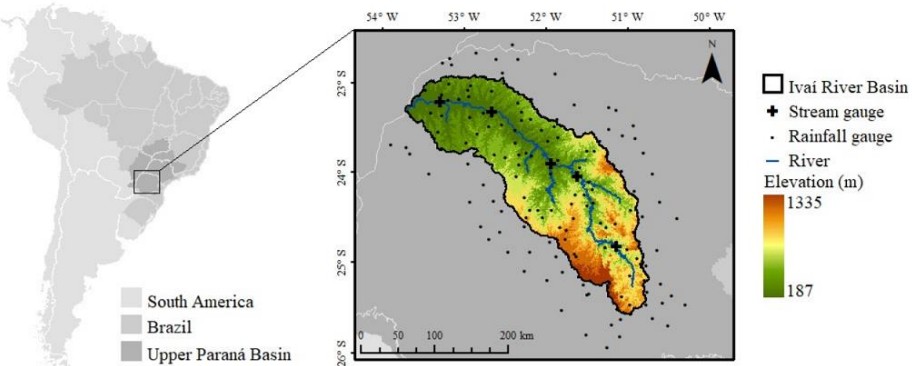

**Figure 1.** Localization map of the Ivaí River Basin within the Upper Paraná River Basin; elevation and rainfall stations are also identified.

The region is characterized by the transitional zone between tropical and subtropical climates. According to the rainfall stations (from 1991 to 2007) specified in Figure 1, the highest precipitation rates occur in the upper segment of the river with 1960 mm and decreases towards the mouth, presenting 1240 mm of the total annual precipitation. Under

the Köppen-Geiger classification [19], the region is characterized by temperate climate with three subgroups: without a dry season, with occurrence of hot and warm summer (Cfa and Cfb), and dry winter with hot summer (Cwa). In general, dry and wet periods are well defined, with the rainy quarter being the December/January/February and the driest June/July/August. As for temperature distribution, the center-south portion of the basin presents mild temperatures with an annual average ranging from 17–18 °C, while the northwestern area presents a mean between 23–24 °C [20].

Approximately 46% of the basin area is used for large-scale agriculture with intensive crop rotation. The remaining area is mainly covered by range grasses, pasture, and forest (15, 14 and 13%, respectively). Only 1.3% are urban areas, although drinking water and industrial supply account for more than 44% of the total water withdrawal. In addition, 75% of the IRB water demand comes from superficial springs and 25% from groundwater extraction [21]. Reservoirs are limited to small installations in the tributaries and there is no water accumulation that affects the natural condition of the river. Such a quasi-natural feature eases hydrological modeling.

### 2.2. The SWAT Model

Soil and Water Assessment Tool (SWAT; [22]) is a watershed scale model developed to predict impacts of water, nutrient, pesticide, and sediment loads under land management practices. It is semi-distributed, physically based, and continuous in time. Different areas of the watershed can be partitioned into topologically connected subbasins. Within the subbasin, Hydrological Response Units (HRUs) are created by a unique combination of land use, soil type, and topography, where most of the hydrological processes are computed and then routed to obtain the total runoff [23]. SWAT aggregates land processes and channel systems for each subbasin composing the watershed behavior. It has been widely used in many applications worldwide and successfully assisted the water-soil-waste nexus [24]. A spatially explicit watershed could close the gap on missing hydrologic information [25]. More information about SWAT is provided on the official model's website (https://swat.tamu.edu/ (accessed on 25 November 2021)).

### 2.2.1. Model Setup

This study was carried out using the 2012 version of SWAT (revision 664). The model was set up for the IRB based on the input data listed in Table 1 and using the ArcGIS interface. Delineation of the subbasins and drainage channels were determined to closely match with the National Water Agency drainage network (in Portuguese: Agência Nacional das Águas e Saneamento Básico—ANA). The watershed was divided into 24 subbasins, with areas ranging from 24.08 to 4416.82 km², detailed in the Supplementary Material, Figure S1 and Table S1. Additionally, the threshold optimization proposed by TopHRU [26] was defined, retaining information of more than 5.0% of the subbasin area for land use, soil type, and slope—totalizing 502 HRUs with 3.4% of the modified basin area. Five streamflow stations were used to assess the streamflow evolution through the channel by selecting the ones placed in the main course of the river. The availability of a common period of daily observations for the five streamflow stations was the decisive factor for choosing the simulation period. Although some of the stations had longer series, the choice of a common period was considered more important, ensuring 17 uninterrupted years of daily observations. The streamflow stations belonged to the monitoring network of the National Water Agency. Another decisive factor was the location—strategically positioned at the limit of morphological and lithological transitions to better stratify drainage areas for the model's assimilation.

**Table 1.** SWAT input data.

| Type | Source | Description |
|---|---|---|
| DEM [1] | SRTM [2] [27] | 90 m resolution |
| Land Use | MODIS [3] | Land use map MCD12Q1 product—500 m resolution, temporal coverage: 2001. Identified Classes: 10. |
| Soil | EMBRAPA [28] and Fauconnier [29] | Soil map 1:50,000 and soil properties. Soil classes identified: 6. |
| Slope | EMBRAPA [28] | Five stratifications: 0–3; 3–8; 8–20; 20–45 and 45–9999 |
| Water use | SEMA [21] | Average hourly water extraction of granted wells and irrigation of large consumers per municipality |
| Climate | CFSR [4] [30]; Global Weather Data for SWAT) | Daily temperature (maximum and minimum; °C), solar radiation ($MJ/m^2/day$), wind speed (m/s) and relative humidity (%). Horizontal resolution ~38 km. Sample points used: 18. |
| Rainfall | ANA–Hidroweb | Daily precipitation derived from interpolation generated from 151 rainfall stations within each subbasin (mm). |
| Streamflow | ANA | Daily readings from five streamflow stations ($m^3/s$) |

[1] DEM—Digital Elevation Model; [2] SRTM—Shuttle Radar Topography Mission; [3] MODIS—Moderate-Resolution Imaging Spectroradiometer; [4] CFSR—Climate Forecast System Reanalysis.

In addition, basic management practices and operations were added to the IRB model, as they could affect the hydrologic balance [23]. This information minimized potential hydrologic disruptions and included particularities of the study area. Among the operating practices, groundwater extraction was added based on the volume of water withdrawn from artesian wells granted by the government [21]. Due to government rules exempting environmental licensing of lowland properties, irrigation was scheduled only for flat terrain. For this reason, it represented 60% of the area among irrigating producers. Sprinkler irrigation is the commonly used method, where water is pumped from streams [31]. For crop rotation (the growth of different crops in succession in one field), two main crops were considered: corn and soybean. In these, each rotation cycle lasts for four years and, every six months, a new growing season begins. The order alternates between corn and soybeans, except for the third year, when corn's interim-harvest assumes the last semester [32]. The last two management practices are exclusive to the agriculture land use classification.

### 2.2.2. Meteorological Input Data

The SWAT model requires five climatic variables to control the water balance and determine components of the hydrologic cycle [23]: precipitation, maximum and minimum temperature, relative humidity, solar radiation, and wind speed. For the last four variables, Climate Forecast System Reanalysis (CFSR); ref. [30] output data from the National Centers for Environmental Prediction (NCEP) were acquired from the Global Weather Data for SWAT website (https://globalweather.tamu.edu/ (accessed on 25 November 2021)). Precipitation datasets were treated following Szcześniak & Piniewski's methodology [33], where readings from rainfall stations were interpolated. Daily behavior of interpolated precipitation was assigned to a single point by the calculation of the average value within the subbasin perimeter and attribution of the same to the respective subbasin centroid. Interpolation was performed by the Inverse Distance Weighting method (IDW). It assumed that the value attribution of an unsampled point was generated by the sum of known points weighted by its inverse distance [34]. As the distance increased, the influence of one data point relative to another declined.

In this study, precipitation stations were taken from the National Water Agency in Brazil. Rainfall data series were chosen based on data availability. Time series were composed of 17 consecutive years, each year with at least 90% of daily readings. After this procedure, a set of 151 rainfall stations were selected (Figure 1). Of these, 83 were situated inside the basin and 68 outside, limited by 50 km from the basin boundary. Based on World

Meteorological Organization (WMO) directives, IRB's inner network was enough to provide data and perform as required at an acceptable cost (inter plains and hilly/undulating surface ranging 575 km$^2$ per station [35]). IRB belongs to one of the areas with the highest density of rainfall stations in South America [36].

### 2.2.3. Parameterization and Sensitivity Analysis

The SWAT model is composed of numerous parameters designed to characterize hydrological processes. The specification of the most appropriate ones is a complex task; fitting parameters is highly conditional regarding the variables used in the objective function. Additionally, direct measurements describing physical systems have limited applicability [37]. SWAT Calibration and Uncertainty Program (SWAT-CUP; version 5.1.6.2, [38]) was applied in the following steps of this study as it enabled sensitivity analysis, calibration, validation, and uncertainty analysis of SWAT model. Uncertainty in the parameters was expressed as intervals and propagated to the output variable, so that model uncertainties could also be evaluated. As the software performed parameter sampling, an envelope of good solutions was created, accounting for the goodness of fit and the uncertainties of the conceptual model.

In addition, parameters can be assigned and calibrated regionally, allowing larger freedom in selecting the distributed parameter scheme [37]. As each section of the IRB consisted of different morphologies and lithologies as identified by Destefani [39], streamflow parameters were regionalized based in three sections: upper (drainage area of Tereza Cristina and Ubá do Sul stations), middle (Vila Rica and Porto Paraíso do Norte stations), and lower (Novo Porto Taquara station and subbasin 4—as shown in the Supplementary Material, Figure S2). Furthermore, IRB modification parameters ranges were selected based on local applications performed for the UPRB. Consequently, the initial amplitude of modifying parameters was physically meaningful and compatible with the study area, parameter definition and modification ranges are shown in Table 2.

### 2.2.4. Model Calibration

Among available techniques for model calibration, SWAT-CUP is a semi-automated model calibration that systematically changes uncertain parameters values and compares model outputs with measured data in natural systems. In this study, we used SWAT-CUP with Sequential Uncertainty Fitting Procedure Version 2 (SUFI-2; [38]). To measure the goodness of fit and the degree that the model accounts for uncertainties, two statistics were used: P-factor and R-factor [38,40]. The P-factor represented the percentage of observed data captured by the envelope, produced by the 95% prediction uncertainty (95PPU), between 2.5% and 97.5% levels of the cumulative distribution of outputs. These were produced by Latin hypercube sampling of the parameter's intervals. The R-factor referred to the thickness of the 95PPU. A balance needed to be reached between both factors; for discharge, Abbaspour [37] suggested 70% of observed data within an envelope of thickness around one. When acceptable values of P-factor and R-factor were reached, parameter ranges reach the desired parameter interval representing model uncertainty.

SUFI-2 performed several iterations aimed at improvement of single simulations based on the objective function. However, no more than four iterations were needed; 500 simulations conducted one iteration for the IRB calibration. Model simulations took place from 1 January 1991 to 31 December 2007, in a monthly time step. The first three years of simulation (1991–1993) were used to warm up the model until it reached a quasi-steady state condition. Calibration was performed for the period between 1994–2005 and validation was performed for the last two years (2006 and 2007). Validation involved running the model using parameters that were determined during the calibration process and comparing its predictions with observed data not used in the calibration [41]. Although validation should represent the entire period of the streamflow recording, limited or nonexistent data or observed data with poor quality were cited among the reasons for short time periods used for studies or for a lack of reported validation. For instance, by 2015

in Brazil, only a quarter of studies published using the SWAT model yielded results for model validation [42]. For the calibration and validation of the IRB, monthly averages of five discharge stations were used in an entire catchment procedure.

**Table 2.** General information about the calibration procedure; parameter name and type of change applied to the parameter sampling, absolute modification range allowed by the model and regionalized modification range applied for the IRB based on areas that drain the UPRB from other studies.

| | Parameter | Description | Modification Range | Regionalized Modification Range |
|---|---|---|---|---|
| 1 | r_ALPHA_BF.gw | Baseflow alpha factor (day$^{-1}$) | 0–1 | ±20% |
| 2 | v_GW_DELAY.gw | Groundwater delay time (days) | 0–500 | 0–120 |
| 3 | v_GW_REVAP.gw | Groundwater "revap" coefficient | 0.02–0.2 | 0.02–0.1 |
| 4 | a_REVAPMN.gw | Threshold depth of water in the shallow aquifer for "revap" or percolation to the deeper aquifer (mm) | 0–5000 | −1000–1000 |
| 5 | v_RCHRG_DP.gw | Deep aquifer percolation fraction | 0–1 | 0–1 |
| 6 | a_GWQMN.gw | Threshold depth of water in the shallow aquifer required for return flow to occur (mm) | 0–5000 | −1000–2000 |
| 7 | v_SURLAG.hru | Surface runoff lag coefficient | 0.05–24 | 0.05–24 |
| 8 | v_ESCO.hru | Soil evaporation compensation factor | 0–1 | 0.65–0.85 |
| 9 | v_EPCO.hru | Plant uptake compensation factor | 0–1 | 0–1 |
| 10 | r_CH_K2.rte | Effective hydraulic conductivity in main channel alluvium (mm/h) | −0.01–500 | ±40% |
| 11 | r_CH_N2.rte | Manning's "n" value for the main channel | −0.01–0.3 | ±20% |
| 12 | r_CN2.mgt | Initial SCS runoff curve for moisture condition II | 35–98 | ±20% |
| 13 | v_CANMX.hru (AGRL/FRSE/RNGE/PAST) | Maximum canopy water storage (mm—discretized for: Agriculture, Forest Evergreen, Range-Grasses and Pasture) | 0–100 | 0–15 |
| 14 | r_OV_N.hru | Manning's "n" value for overland flow | 0.01–30 | ±30% |
| 15 | r_SOL_AWC().sol | Available water capacity of the soil layer (mm H$_2$O/mm soil) | 0–1 | ±20% |

Parameters can be aggregated based on hydrological group specification, and are formulated as it follows: x__<parname>.<ext>__<hydropraph>__<soltex>__<landuse>__<subbsn>__<slope>, where x_ is the code that indicates the type of change applied to the parameter, v_ means the existing parameter value is to be replaced by a given value, a_ indicates that a given value is added to the existing parameter value, r_ indicates that an existing parameter value is multiplied by (1+ a given value), <parname> references the SWAT parameter name, <ext> is the file extension code, <hydrogrp> references soil hydrological group, <soltext> is soil texture, and <landuse>, <subbbsn> and <slope> refer to land use category, subbasin number, and slope, respectively.

### 2.2.5. Model Evaluation

Single simulation performance was evaluated by statistical analysis by measuring the strength of a linear relationship between simulated and observed data [43]. Nash Sutcliffe Efficiency (NSE, [44]) was the objective function to compare the performance of individual simulations using observations as reference. The NSE values ranging between 0.65 and 0.75 were considered good. Values greater than 0.75 were considered very good [43]. However, an assessment of the absolute or volume error in combination with efficiency criteria to complement the model's reliability was recommended [45]. The second statistic was RSR, which was the RMSE-observations standard deviation ratio. RSR standardized RMSE (Root Mean Square Error) using the observations standard deviation. The last statistic rating was the percent bias (PBIAS), measuring the average tendency of simulated data to be larger or smaller than their corresponding observations [46]. The optimal value was

0; positive values indicated model underestimation bias and negative values indicated model overestimation.

### 2.3. Flow Duration Analysis

In addition to the model's performance ratings and statistical techniques, observed and estimated streamflow measurements were evaluated under the flow threshold indices. Those indices have been extensively applied in water resources planning [47]. Flow duration curve (FDC) was used to demonstrate the relationship between flow and frequency in which this flow was equaled or exceeded, as a cumulative distribution function [48]. In an overall interpretation, the FDC evaluates the nature of the flow regime and the conditions in the river channel. A steeper flow duration curve indicates flashy flow regimes with little baseflow and large storm hydrographs, and a shallow curve indicates sustained groundwater flow with minor stormflow [49].

Some percentages of exceedance piqued greater interest during the hydrological analysis. For example, the flow rate of 50% of exceedance of time corresponded to the median flow, identified by Q50. The values of Q10 and Q90 represented the flow rate, which was exceeded by 10% and 90% of the time, respectively. Q90 is often used as a low flow rate [50]. The Q90/Q50 ratio was applied as a baseline contribution indicator [51], whereas Q10/Q50 indicated flood peaks [52]. Q10/Q90 was employed as an indicator of flow variability [53] and Q95 was used to define the operability of firm power generation from hydroelectric plants [54], in addition to being the minimum flow to protect the river [55].

## 3. Results

### 3.1. Model Assessment

The general performance of the calibrated SWAT model for the IRB showed good results (see Supplementary Material, Table S2 for statistics). Sensitive parameters used during calibration as their final modification range and fitted value are shown in Table 3. After two iterations, calibration of monthly discharge revealed very good results ($0.75 < \text{NSE} \leq 1.00$) for four streamflow stations and good performance for one ($0.65 < \text{NSE} \leq 0.75$) [43]. On the headwater, Upper Ivaí, the 95PPU interval captured 70% and 73% of the observed data for Tereza Cristina and Ubá do Sul stations, respectively, with an R-factor of 0.72 and 0.83. In Middle Ivaí, the index captured 67% of the total data for the Vila Rica station with a thickness of 0.49, whereas Porto Paraíso do Norte performed better with thickness around 1.00 capturing 92% of observed readings. For Novo Porto Taquara (Lower Ivaí), the monitoring station with greater drainage area, the envelope bracketed 83% of observed data. Further discussion of individual performance of results for each monitoring station can be found in the following section.

Based on PBIAS evaluation of the best simulation ($n = 129$), the average trend of streamflow estimates showed only negative values for the Porto Paraíso do Norte, which exhibited a 3.9% deviation in the simulated data, indicating the overestimation of the streamflow by the calibrated model. As for the positive values, which revealed the underestimation of streamflow, in ascending order, we obtained: 6.0% for Ubá do Sul; 6.7% for Tereza Cristina; 8.8% for Novo Porto Taquara, designating very good performance rating (PBIAS < ±10); and satisfactory to Vila Rica with 16% of PBIAS, for a satisfactory rating($\pm 15 \leq$ PBIAS < ±25) [43]. For the RSR, which incorporated the error rate of the simulated values—the closer to zero, the better the model performance—we identified the following in ascending order: Tereza Cristina with 0.33; Porto Paraíso do Norte presenting 0.35; 0.36 for Novo Porto Taquara; 0.37 for Ubá do Sul; and, with greater error, Vila Rica with 0.55. Even though it presented with the highest RSR, the performance of Vila Rica station was still considered good ($0.50 < \text{RSR} \leq 0.60$).

**Table 3.** Parameters used during calibration on SWAT-CUP and its rank based on sensitivity analysis. Initial values during model setup, final modification range from the best iteration and best modification factor, and remaining modification interval (RMI) regarding the best simulation are also described.

| | Parameter | Rank | Segment | Subbasin | Default Value | Final Modification Range | RMI * (%) | Best Modification Factor |
|---|---|---|---|---|---|---|---|---|
| 1 | r_ALPHA_BF.gw | 13 | Upper | 24 | 0.093 | −0.027–0.317 | 86.0 | 0.288 |
| | | | | 16, 17, 19–23 | 0.0670 | −0.027–0.317 | 86.0 | 0.288 |
| 2 | v_GW_DELAY.gw | 22 | Lower | 1–5, 8 | 31 | −31.751–69.431 | 84.3 | −8.783 |
| | | 17 | Middle | 6, 7, 9–15,18 | 31 | 3.167–81.072 | 64.9 | 55.909 |
| | | 1 | Upper | 16, 17, 19–24 | 31 | −55.514–61.514 | 97.5 | −22.160 |
| 3 | v_GW_REVAP.gw | 24 | Lower | 1–5, 8 | 0.020 | 0.011–0.070 | 32.7 | 0.067 |
| | | 20 | Middle | 6, 7, 9–15, 18 | 0.020 | 0.010–0.067 | 36.6 | 0.010 |
| | | 11 | Upper | 16, 17, 19–24 | 0.020 | 0.041–0.083 | 23.3 | 0.045 |
| 4 | v_RCHRG_DP.gw | 26 | Lower | 1–5, 8 | 0.050 | −0.195–0.601 | 79.6 | 0.004 |
| | | 14 | Middle | 6, 7, 9–15, 18 | 0.050 | 0.294–0.883 | 58.9 | 0.303 |
| | | 3 | Upper | 16, 17, 19–24 | 0.050 | −0.369–0.543 | 91.2 | −0.200 |
| 5 | a_GWQMN.gw | 19 | Lower | 1–5, 8 | 1000 | −140.800–1578.800 | 57.3 | 562.516 |
| | | 21 | Middle | 6, 7, 9–15, 18 | 1000 | −2405.806–531.806 | 97.9 | −1803.595 |
| | | 5 | Upper | 16, 17, 19–24 | 1000 | −89.810–1731.810 | 60.7 | 1764.520 |
| 6 | v_SURLAG.hru | 7 | Middle | 6, 7, 9–15, 18 | 2 | −4.347–14.554 | 78.9 | −0.434 |
| 7 | v_ESCO.hru | 9 | Lower | 1–5, 8 | −0.095 | 0.553–0.751 | 99.0 | 0.640 |
| | | 8 | Middle | 6, 7, 9–15, 18 | 0.095 | 0.707–0.823 | 58.0 | 0.774 |
| | | 6 | Upper | 16, 17, 19–24 | 0.095 | 0.594–0.764 | 85.0 | 0.745 |
| 8 | v_EPCO.hru | 23 | Lower | 1–5, 8 | 1 | 0.441–1.324 | 88.3 | 1.056 |
| | | 25 | Middle | 6, 7, 9–15, 18 | 1 | 0.412–1.237 | 82.5 | 1.230 |
| | | 28 | Upper | 16, 17, 19–24 | 1 | −0.123–0.625 | 74.8 | 0.400 |
| 9 | r_CN2.mgt | 27 | Lower | 1–5, 8 | Variable | −0.139–0.087 | 56.5 | −0.069 |
| | | 2 | Middle | 6, 7, 9–15, 18 | Variable | −0.003–0.389 | 98.0 | 0.267 |
| | | 4 | Upper | 16, 17, 19–24 | Variable | −0.197–0.067 | 66.0 | −0.023 |
| 10 | v_CANMX.hru (AGRL) | 29 | Middle | 6, 7, 9–15, 18 | 0 | 3.500–11.169 | 51.1 | 8.968 |
| | | 15 | Upper | 16, 17, 19–24 | 0 | 5.930–17.799 | 79.1 | 16.125 |
| | v_CANMX.hru (FRSE) | 16 | Upper | 16, 17, 19–24 | 0 | 6.021–18.068 | 80.3 | 11.045 |
| 11 | r_SOL_AWC().sol | 18 | Lower | 1–5, 8 | Variable | −0.033–0.301 | 83.5 | −0.028 |
| | | 12 | Middle | 6, 7, 9–15, 18 | Variable | −0.221–0.059 | 70.0 | −0.184 |
| | | 10 | Upper | 16, 17, 19–24 | Variable | −0.237–0.054 | 72.7 | −0.184 |

* RMI, remaining modification interval is the ratio between the final calibrated range from the best iteration and the modification range allowed by the model.

### 3.2. Flow Duration Curve Assessment

Each flow contribution at each monitoring station is illustrated in Figure 2. We verified that the years of 1997 and 1998 stood out for high values of flow. Following those years, a period of drought was identified at the end of 1999, when one of the more severe droughts of the historical series occurred. These events were associated with the episodes of El Niño (1997/1998) and La Niña (1999/2000), which were characterized by anomalies in the sea surface temperature of the Equatorial Pacific. Such anomalies disrupted global circulation and produced significant impacts on precipitation in several regions of South America. In general, studies have indicated that the region in which IRB is located may be influenced

by both phases of the oscillation, with positive (El Niño) and negative (La Niña) impacts on precipitation [56–58].

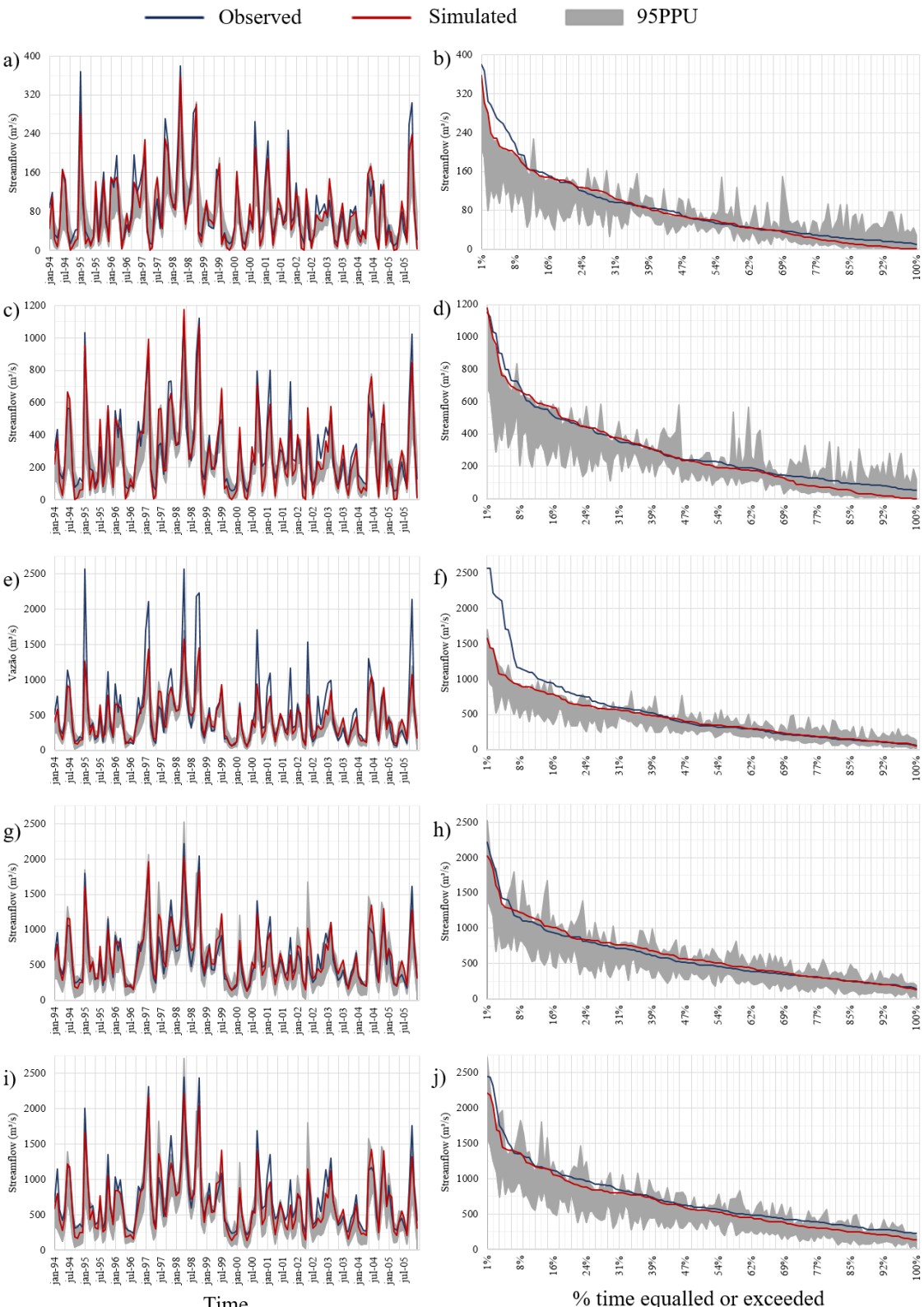

**Figure 2.** Simulated and observed hydrographs and respective flow duration curves for every monitoring station from IRB. From top to bottom: Tereza Cristina (**a**,**b**), Ubá do Sul (**c**,**d**), Vila Rica (**e**,**f**), Porto Paraíso do Norte (**g**,**h**), and Novo Porto Taquara (**i**,**j**).

In relation to the other years of the simulation, the Ivaí River did not have a defined seasonal behavior. Events of flood and drought were indeterminate; thus, the flow regime was configured with low frequency. Hydrographs showed pointed peaks close to each other, referring to rivers with structured slopes with rock and impermeable soils. Furthermore, precipitation generated flows with acute and rapid peaks flowing through the river channel [59].

In order to understand the discrepancies between simulated and observed flow regimes, each station was treated individually, aiming to identify particularities of the basins' drainages and the performance of the model. The 95PPU intervals from calibration are shown in Figure 2. To present the uncertainty within the FDC, the interval was constructed respecting the uncertainty of each simulated monthly flow. Each station was discussed in the following sections, and results are summarized in the Supplementary Material, Table S3.

### 3.2.1. Tereza Cristina

Tereza Cristina is the highest station in the main course of the river and features the smaller coverage area, 3572 km$^2$. It presents a complex drainage pattern with embedded riverbed, creating rapid surface runoff and low baseflow [39]. The modeled average flow rate for the historical series of 1994–2005 was 80.18 m$^3$/s, slightly smaller than the observed rate. This was identified by the difficulty of the model in reproducing the flow peaks in this segment. Even though the behaviors of the peaks were aligned, the magnitudes were, in most cases, smaller. Underestimation was also seen on baseflow, wherein minimum flows were not sustained (Figure 2a).

As mentioned before, the steepness of the FDC illustrated streamflow variability. Comparing curvatures between Tereza Cristina simulation and observation, in the first ten exceedance percentiles, the dissimilarity among curves indicated the limitation of the calibrated model in reproducing maximum flows. This pattern also repeated for minimum flows, in which the curves distanced themselves as simulation dried the river for higher percentages. Under the Q90/Q50 ratio, indicating the groundwater contribution, we found 0.30 for the observed and 0.11 for the simulation. For values closer to zero, the river stopped flowing for rates below 90% of the time (Figure 2b).

The 95PPU, highlighted in grey in Figure 2, represented the thickness of the envelope, which could range from zero to infinity. Values around 1 were acceptable and indicated an envelope with the same width of the standard deviation of the corresponding observed variable [60]. The Tereza Cristina's model uncertainty used 72% of the standard deviation thickness but was not able to track high flow behavior as the best simulation (in red). It was on the upper edge of the uncertainty envelope (Figure 2a). The regional character of the relief and the complex drainage pattern were considered as factors that could have hindered the representation of physical behavior in this portion of the IRB. The envelope feature of Figure 2b, with increasing frequency, indicated a narrowing envelope for flows between 30 and 70%, followed by a thickening of the 95PPU for low flows, limited by the axis to zero.

### 3.2.2. Ubá do Sul

The exclusive drainage portion of Ubá do Sul comprised subbasins 16, 17 and 19–23. As for the FDC, the maximum flow values were well represented. However, the simulated curve declined in a concave shape while the observed decreased in the form of steps. At the second crossing of the curves, approximately 18% in the FDC (Figure 2d), the behavior of both curves resembled until Q50, when the model underestimated streamflow once again. For Q90, the low flow indicator, the simulation obtained 23.96 m$^3$/s, which was three times smaller than observation (83.05 m$^3$/s), this discrepancy repeated itself for Q90/Q50 ratio. This behavior in Ubá do Sul resembled the Tereza Cristina station, indicating poor performance of the model for a higher percentage of exceedance for the Upper Ivaí. The drainage area of Tereza Cristina added to Ubá do Sul completed the Upper Ivaí territory.

Part of the model's difficulty in representing low flows was attributed to the transitional zone of two hydro-stratigraphic units of the Paraná Sedimentary Basin, the Serra Geral Aquifer System, and the Guarani Aquifer. The transitional zone implied the maintenance of the base level of the rivers and affected the discharge patterns between the two aquifers along with the recharge zones [61]. This resulted in great complexity of the hydrological variables and groundwater components of the region. Furthermore, SWAT did not represent the physical behavior of this transition zone; therefore, most of the baseflow at this station was underestimated, which compromised the model's performance.

For the 95PPU of Figure 2d, the concave shape of the envelope resembled the observed and simulated curvature, a characteristic of representing the flow variability indicating better representation of the features of the basin. In terms of thickness, the envelope rated 0.83, although it still showed some difficulty in representing the high flow rates. This was an expected behavior since this station and Tereza Cristina are part of the same regionalization of parameters and possess the same physical characteristics.

### 3.2.3. Vila Rica

Vila Rica drains the highlands of the Middle Ivaí, comprising the subbasins 11–13, 15, and 18. The observed mean flow was 533.64 m$^3$/s, and for simulation, it was 448.14 m$^3$/s. Regarding the standard deviation of the observed flow, it reached 504.3 m$^3$/s while the simulated did not exceed 306.84 m$^3$/s, indicating an underestimation of the amplitude of variation for the general flow behavior (Figure 2e). The calibrated model presented difficulties in simulating the maximum flows with quality and, although the subbasin responses portrayed the flow behavior, the intensity was inadequate. However, analyzing the FDC of Vila Rica, the model's curve met the natural behavior from 30% onward; that is to say, 70% of the behavior was portrayed by the model. Additionally, the discrepancies attributed to the model's performance consisted of the first 30 percentiles of exceedance.

The probable explanation for the difficulty in simulating higher flows was attributed to the particularities of the section between Ubá do Sul and Vila Rica regarding the flow contribution. The highest contributions per unit of area derived from the drainage between Ubá do Sul and Vila Rica. For this section of the basin, the contribution was about 43% of the flow observed in Vila Rica, while the drainage area corresponded to about 34%. In Leli et al. [62], the authors investigated the flow contribution rate of the main tributaries of IRB, demonstrating that, in the Vila Rica-Ubá do Sul domain, the contribution was extremely high. The same authors concluded that the contribution along the river derived primarily from small tributaries. Apart from that, to explain the anomaly of the high flow contribution of restricted small areas, Paiva [63] suggested that this region was heavily influenced by groundwater transfer due to local distribution and concentration of fractures in the basaltic soil. In addition, Bittencourt et al. [64] suggested that the recharge of the underground reservoir consisted of rainfall added by water that ascended from geological discontinuities of underlying aquifers under high hydraulic pressure. To be able to portray this addition of groundwater to the system, the SWAT would need to gather new basin inlet components—manually and with assigned values. These data are not easily measured and were not available for the study region. As such, the groundwater transfers described by Paiva [63] and Bittencourt et al. [64] were not included in the model. Therefore, future work may include improvements in this direction.

Nevertheless, it was from Vila Rica that the model captured the baseflow behavior, which contributed to the perennialism of the Ivaí River. From the perspective of the groundwater ratio contribution (Q90/Q50), it matched 0.341 for simulation and 0.359 for observation. This could also be identified in the overlapping curves decaying together (Figure 2f). For the extreme low flow condition indicator (Q95), the simulation obtained 94.94 m$^3$/s and observation obtained 88.38 m$^3$/s, meaning that the calibrated model ensured minimum flow estimation from this station. This behavior was also seen with the 95PPU envelope, which did not touch the zero axis when the river was no longer considered perennial. Throughout the envelope representation, the thickness remained

narrow (0.49), and, although it did not capture flow rates greater than 500 m$^3$/s, the behavior for smaller flows proved to be quite representative. The model upstream had not captured this behavior until then.

### 3.2.4. Porto Paraíso do Norte

The exclusive drainage area at the station Porto Paraíso do Norte, subbasins 6–7, 9–10 and 14, comprised the boundary between the Serra Geral and Caiuá formations, which featured a smoothly undulated relief [65]. The area was also in a land cover transitional zone due to the soil fertility. Upstream, the basaltic formation permitted intensive agriculture, while downstream, a coarser sand texture was observed, ideal for a crop and livestock system. Thus, the positioning of Porto Paraíso do Norte is strategic for the delimitation of the Middle Ivaí.

In this station, the 95PPU exceeded the maximum flow for the first time (Figure 2g), specifically in the year 1998, which was underestimated in upstream stations. The simulations for Porto Paraíso do Norte presented the best performance, as the model and its uncertainties bracketed the observed readings, 92%. Another interesting pattern is that the envelope thickness decreased with decreasing flow magnitude, indicating some proportionality of uncertainty. Furthermore, the curvature of the 95PPU and best simulation resembled the observation as well. For the best simulation, the standard deviation differed 10 m$^3$/s less in relation to the observation.

The FDC of streamflow (Figure 2h) presented a lag that was shortened twice as two crossovers occurred in 7 and 20%, causing the inversion behavior among curves. Although the difference among curves was persistent, the flow variability, measured by the ratio Q10/Q90, was minor: 5.26 and 5.23 for simulated and observed, respectively. This rate proximity indicated that the model reproduced the overall flow regime. Furthermore, the curves approach themselves until they overlapped from a 70% exceedance rating, demonstrating the good performance of the low flow simulation. This was also observed in the Q90/Q50 ratio. The Q95 of the ensured electric energy production was 188.69 m$^3$/s for the simulated flow and 196.96 m$^3$/s for the observed, which was twice the upstream station's flow magnitude. As identified at the Vila Rica station, the model calibration offered conditions to reproduce low flow predictions.

### 3.2.5. Novo Porto Taquara

Novo Porto Taquara drains 34,432 km$^2$, representing 94.1% of the entire IRB. The station is positioned at 240 m above sea level and 65 km from the river exuter. The exclusive area of Novo Porto Taquara (subbasins 1–3, 5 and 8) added to the drainage area to the mouth (subbasin 4) determines the extension of the Lower Ivaí. This extension coincides with the development of the alluvial plain, traversing 150 km to the mouth [66]. The Ivaí river discharge interacts with the changes in fluvial morphology by increasing the width of the central course, margins, and water transposition on dikes [39]. Closer to the mouth, a flow reduction occurs due to the low flow velocity caused by the Paraná River's impoundment [67]. Thus, this region was parameterized to respect the alluvial plain, which also develops in a drainage area with the low slope surrounded by outcrops of the Caiuá Sandstone.

Note that the simulated flow hydrograph was below observation records for maximum and minimum values, as shown in Figure 2i. Furthermore, for the FDC, higher discrepancies occurred at the maximum flows, where the difference reached more than 300 m$^3$/s (approximately in 2%), with a lag of about 70 m$^3$/s to other flows and a narrowing between 30% and 60%. These underestimations were demonstrated by the ratio Q10/Q90, which differentiated among discharges, increasing with respect to observation. The average monthly flow obtained was 655.04 m$^3$/s, or 62.84 m$^3$/s less than the observation, indicating that the model again underestimated the simulated flow rate. Regarding the uncertainties of the model, the behavior of the 95PPU interval resembled that of the streamflow station upstream. This was a very positive feature since there is a strong transition of basin charac-

teristics between the Middle and Lower Ivaí. Although the thickness was reduced to 0.91, and fewer data have been enveloped (a decrease of 9%), the results were still very good.

### 3.3. Model Validation

Model validation for the IRB was performed for the years 2006 and 2007, predominantly a dry period, registering monthly averages lower than the calibration interval (1994–2005). This behavior was attributed to rainfall anomalies in the region, which produced lower monthly flows during the first months of 2006 and later, in the last half of the same year when they reached maturity, prolonging the effects until the beginning of 2007 [68]. While the procedure was to reserve at least one-third of the simulated period for validation, the directive was also to maximize data assimilation so that all behavior could be assimilated by the model during calibration. In this sense, as the ENSO events, isolated or in combination with opposite phases, were distributed throughout the available data period, breaking the rule should only exert a small or weak effect on model performance. In order to include a period without ENSO episodes, the calibration period was extended to the year 2005, at the cost of the reduction of the validation period. Although the remaining period for validation was shorter than the recommendations called for, it represented the calibration period, as it included a weak El Nino and a strong La Nina. The 95PPU managed to cover 88% of the observational records for Novo Porto Taquara (closest to the mouth) and Tereza Cristina (positioned at the head). Down the river, Ubá do Sul and Vila Rica stations surpassed the calibration performance, enveloping 92% of the readings. Porto Paraíso do Norte, with the best P-factor, reached 96% of the observed records. The R-factor maintained satisfactory values for all stations, especially Vila Rica, which obtained very good indexes. About 88% of the streamflow data from the validation period were enveloped, with a thickness of 1.25.

Regarding the statistical indices for the best simulation of the validation (*n* = 18), in general, the performance of the model for each monitoring station approached that of the calibration. However, this did not include Vila Rica, which, with an NSE index of 0.83, was enhanced from Good (in calibration) to Very Good (0.75 < NSE $\leq$ 1.00). The same quality of performance was observed for PBIAS (PBIAS < $\pm$10) and for RSR (0.00 $\leq$ RSR $\leq$ 0.50). The NSE obtained for the other stations, when compared to the calibration, suggested a small reduction in the quality of the simulation as they presented slightly smaller indices. The same occurred with the RSR index; however, for the Ubá do Sul station, the value 0.88 remained the same for the calibration and validation. The PBIAS of Porto Paraíso do Norte remained negative, indicating a persistent overestimation of the flows. For the other stations, PBIAS maintained positive values, as well as calibration. However, Tereza Cristina, Ubá do Sul, and Novo Porto Taquara were downgraded to good performance ratings.

### 4. Discussion

In a preliminary investigation, physiographic and climatic measurements were assembled with hydrological modeling to better understand cumulative effects on the basin. The application of the SWAT model to the IRB was a preliminary effort to describe the monthly volume of water, i.e., river discharge. With that said, there is no single best model, since numeric models and computation performances can be continuously improved and refined. Additionally, changes and alterations in the flow regime cannot be predicted until the flow is analyzed [69].

Once again, using the FDC, streamflow measurements, concerning simulated and observed data, were examined, aimed at model functionality analysis regarding flow criteria. Herein, we used the volume error calculated for each individual exceedance percentage. This procedure allowed us to detect the errors pertaining to each parcel of the streamflow exceedance percentage. The procedure was replicated to each monitoring station, as shown in Figure 3. For streamflow in monthly and annual time steps, Donigian et al. [70] stated that the absolute value of the volume error for individual fits were considered very good when it was lower than 10%, good when the error was between 10% and 15% and fair for

errors between 15% and 25%. Singh et al. [71] adopted the same rating in SWAT simulations, whereas, for the other author, the rating was referred to simulations using the Hydrological Simulation Program FORTRAN.

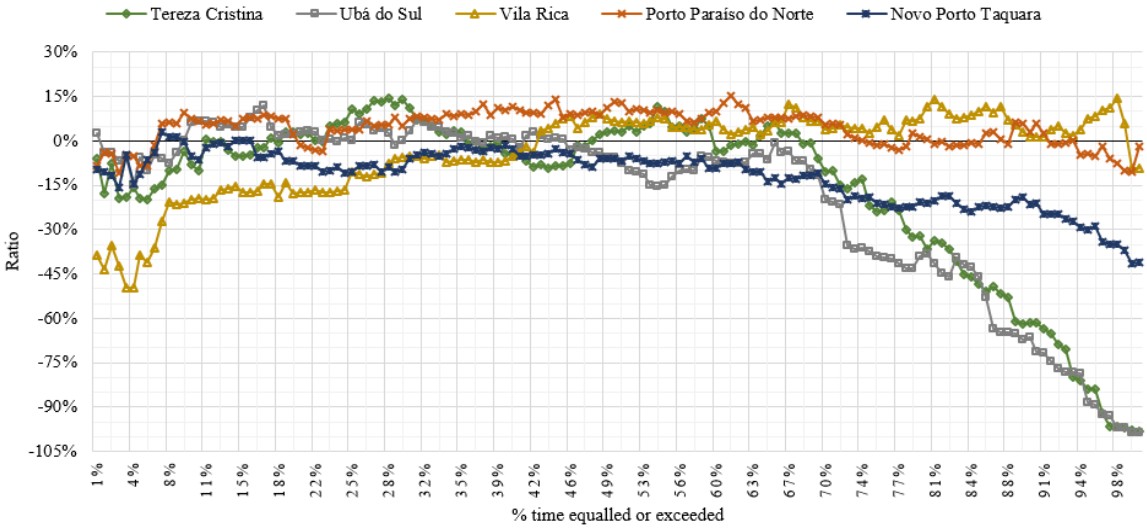

**Figure 3.** Volume error of flow duration curves.

When examining the stations' overall behavior, only Porto Paraíso do Norte presented results inside the good rate coverage, since the absolute maximum percentage error did not exceed 15%. Additionally, this result aligned its performance with the statistical indices presented earlier. To our understanding, the calibration results for this monitoring station produced reliable interpretations for the whole FDC. Thus, its predictions could be adopted by Ivaí River managers with minimum reservation.

Vila Rica high flows held great errors at the beginning of the FDC, ranging up to −49% of error, where the observed streamflow reached 2142 m$^3$/s and the simulation reached merely 1077 m$^3$/s. Fair results for this station were found in the interval between 7–13% of the FDC. However, for the following 10%, values fluctuated between good and fair rates. From 25% on, the behavior sustained good results, with an inversion of error in 43% and 99%.

For the other stations, the good performance rating was constant until the breaking point in about 70% of the FDCs, in the convergence of the curves. From then on, the behaviors differed; Novo Porto Taquara descended slowly, while the stations of Upper Ivaí decayed rapidly. The fair attribution was lost at the FDC value of 93% for Novo Porto Taquara, the most representative station, and at 72% and 77% for Ubá do Sul and Tereza Cristina, respectively. Up to these limit values, the model should be used with caution. For higher percentages of exceedance, the model must be neglected until further improvements. When performance rates reach acceptable minimum rates, the model will be able to assist low flow forecasting.

Our application of the well-established SWAT model proved to be a relevant tool to describe a framework for Ivaí hydrology. From an operational perspective, hydrological models should expose the limitations and credibility of their responses so that managers, when they take ownership of this tool, are aware of its usefulness and take advantage of its functionality. As for the future, this model could be used to address a range of questions including infrastructure projects, land cover substitution, and shifts in meteorological and hydrological conditions. Nevertheless, simulations could be integrated with sediment, nutrients, and water quality modeling since SWAT is a high complexity model.

Among the characteristics of the IRB, the calibrated model presented shortcomings in representing the complex set of underground structures. Measurement records that characterize these components are not common and are limited to shallow and sparse

sampling locations. However, once reasonable measurements are available, the model could be restructured with contribution plots in the water balance equation and support better results as physical behavior is better assimilated. This external intervention could prove useful, particularly in the Vila Rica station wherein the calibrated parameters of Middle Ivaí were not able to capture the physical behavior of the fractures in the basalt, which added water to the system, producing maximum peaks of streamflow.

Nevertheless, this model application contributed to the description of the Ivaí River behavior in its non-dammed course and could prove valuable to several projects as hydroelectric plants are planned to bar the main course of the river. A better knowledge of the current non-dammed configuration could mitigate possible negative scenarios and become the basis for decision making for ongoing hydrology. Our approach could avoid unwanted consequences and allocate opportunities associated with the inclusion of this new framework.

## 5. Conclusions

In this study, we set and calibrated the widely used hydrological model SWAT for monthly discharge in the Ivaí watershed, a subbasin of the Upper Paraná River basin. The application of the model and verification of its operational adequacy ensured that this tool was used with due attribution. Our analyses supported the position that the totality of information should be handed over to managers, and where it should be used sparingly. Furthermore, they could indicate directions in which the collection of data that are more representative of physical and management processes could help improve the proposed models, making them even more robust and better able to reproduce conditions of public interest. Concerning the calibration procedure and operational adequacy, we were able to highlight the following conclusions:

1.  The calibrated SWAT model was suitable for use in addressing a range of questions, including infrastructure projects, land cover substitution, transport of sediment, nutrients, and assessment of water quality, as well as shifts in meteorological and hydrological conditions due to climate variability or change.
2.  Calibrated parameters were not able to capture the effects of the fractures in the basalt that added water to the streams around the Middle Ivaí River. This effect was observed mainly in the form of maximum peaks of streamflow.
3.  Under the operational use perspective, the performance of the calibrated model presented in this study, proven by several statistical indexes, suggested that SWAT predictions could be broadly adopted by Ivaí River managers with minimum reservation, albeit only under extreme streamflow conditions.
4.  The Ivaí River is a non-dammed course—but several projects involving hydroelectric plants have been planned are advancing into their implementation phases at the time of this study. Good calibration in the current configuration could build up a broad range of opportunities to investigate future scenarios and avoid unintended consequences.

To our understanding, in Brazil, the knowledge bases in hydrological modeling and water resource management have a gap, largely because information is poorly transferred. Most simulation outcomes are not adequately explicit or are strictly theoretical concepts that do not cope with adequate communication. This has led to a situation in which both sides, i.e., scientists and managers, are impaired. Consequently, the acquired knowledge from models has been considered speculative and have not been applied as an instrument to improve river basin planning, regulation, and management actions. However, several countries count on hydrological modeling to provide input to draw public policies and facilitate consistent planning. There have been several national and continental recommendations that were successfully based on the model's directives. Thus, proposing a deterministic model with a physical basis interpreted for control points along the river could help others to understand the basin based on descriptive characteristics that are not moderated by a single sector of the economy. Finally, we expect our contribution to help bridge the gap between scientists, policymakers, and managers in water resource management.

**Supplementary Materials:** The following supporting information can be downloaded at: https://www.mdpi.com/article/10.3390/w14030488/s1, Figure S1: Identification of delineated subbasins of Ivaí River Basin. Figure S2: Ivaí River main areas used to regionalize parameters during calibration. Upper Ivaí River in red drains; Tereza Cristina and Ubá do Sul streamflow stations; Middle Ivaí in blue drains; Vila Rica, Porto Paraíso do Norte; and Lower Ivaí in green draining; Novo Porto Taquara and subbasin 4. Table S1: Detailing of the Ivaí Rivers subbasins with number and name of the main tributary. Area, altitude, and number of hydrologic response units per subbasin are also identified. Table S2: Summary statistics for calibration and validation (inside brackets) periods of the SWAT modeling for the five fluviometric stations of IRB model. Table S3: Flow threshold indices and References

**Author Contributions:** Conceptualization, T.F., M.V.B.d.M., E.D.d.F. and J.A.M.; Data curation, T.F., M.V.B.d.M., A.P.R., M.M.d.E., A.C.F.X., S.A.A.R. and E.B.S.; Formal analysis, T.F., M.V.B.d.M., V.C.D.S. and J.A.M.; Investigation, T.F. and M.V.B.d.M.; Methodology, T.F., M.V.B.d.M. and V.C.D.S.; Software, T.F., M.V.B.d.M., V.C.D.S. and S.A.A.R.; Supervision, M.V.B.d.M., C.B.U., R.A.F.d.S., E.D.d.F. and J.A.M.; Writing—original draft, T.F.; Writing—review & editing, T.F., M.V.B.d.M., V.C.D.S., S.A.A.R., L.D.M., C.B.U., E.D.d.F. and J.A.M. All authors have read and agreed to the published version of the manuscript.

**Funding:** This research was funded by Coordination for the Improvement of Higher Education Personnel (CAPES Coordenação de Aperfeiçoamento de Pessoal de Nível Superior), Process nº 88887.115875/2015-01 and CAPES PROEX 0344/2021, and by National Council for Scientific and Technological Development (CNPq—Conselho Nacional de Desenvolvimento Científico e Tecnológico) Process nº 140033/2020-3.

**Institutional Review Board Statement:** Not applicable.

**Informed Consent Statement:** Not applicable.

**Acknowledgments:** The authors would like to gratefully acknowledge the Brazilian National Water Agency (ANA) for providing the precipitation and streamflow data. Also thank José Alberto Fernandez Monteiro for his comments and discussion which helped improving the quality of the manuscript.

**Conflicts of Interest:** The authors declare no conflict of interest.

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
