# Peer review of "Simulating Discharge in a Non-Dammed River of Southeastern South America Using SWAT Model"

_water, doi:10.3390/w14030488_

Round 1

Reviewer 1 Report

The study undertaken by the authors is not distinguished by originality but presents a careful and in-depth analysis of the subject approached. The elements of the natural setting are so carefully and in detail presented that the reader from any corner of the world could not feel a stranger to Brazil.

The processing of hydrological data is done in detail and professionally. The results obtained from the simulations are interpreted correctly and can really be the basic standard tool for planning the water resources in the studied area.

The studied bibliography can always be a starting point for a similar scientific investigation.

In conclusion, the manuscript is well written and can be interesting for specialists in the field of river basin management. I propose its publication in this current form.

Author Response

We appreciate the reviewer comments and the time dedicated to our manuscript. We have included some notes to help them further support the conclusion. All changes were highlighted in green color in the document. 

We highlight the objectives of the study in the discussion and conclusion parts that were requested by the editor. The inclusions are indicated below.

Line 545: We changed “The proposed IRB model is a preliminary effort to describe…” to “The    application of SWAT model to the IRB is a preliminary effort to describe the monthly volume of water, as river discharge.”

Line 609: We added the following sentences to the conclusion “The application of the model and verification of its operational adequacy ensures that this tool is used with due attribution. Such analysis helps to state that the totality of information is handed over to managers and where it should be used sparingly. Furthermore, they can indicate directions in which the collection of information that is more representative of physical and management   processes can help improve the proposed models so that they are even more robust and better reproduce the conditions of public interest and become adequate at each update.”

Reviewer 2 Report

I just want to thank the authors for their work on this manuscript. They justified the purpose of the work by showing how it does feel a great need for the hydrological community in Brazil. I especially liked the supplementary material which gave much more insight to how the SWAT model was put to use.

My suggestions are editorial in nature:

  • Line 22 (in abstract) - there should be a comma between "Ivaí River Basin" and "an".
  • Lines 34-35 - the comma should be placed after "world" and not after "and".
  • Line 52 - is it only one plant or are there multiple plants in question?
  • Line 67 - the "to" in "this was not the case to the URPB" probably should be a "for".
  • Line 83 - "Despite those progress" should be changed to "Despite that progress".
  • Lines 85 & 86 - "And this is evidenced" -- I would cross out the "And" and start the sentence with "This is evidenced."
  • Line 441 - I would change it to "These data are not easily measured" (sorry, it's a pet peeve of mine as my students would probably tell you)
  • Lines 506 & 507 - It should probably be "less data have been enveloped."
  • Line 521 - the "that" in the beginning of that line should be "than".

Author Response

We would like to thank the reviewer for his comments and time spent reading our manuscript. All changes were highlighted in green color in the document. The intention to create supplementary material describing how the model was organized is one of the characteristics that we would like to expose since they are not always available in this way. In that sense, we are glad to hear that the reviewer has verified and appreciated this. For works like this, detailed and peer-reviewed, we believe that water resources managers should have access to inform decision-making. We are very grateful.

Thanks for the corrections in the text, that certainly improved the document.

  • Line 22 (in abstract) - there should be a comma between "Ivaí River Basin" and "an".

Done.

  • Lines 34-35 - the comma should be placed after "world" and not after "and".

Done.

  • Line 52 - is it only one plant or are there multiple plants in question?

There are several plants in the region. To avoid any doubts, we changed the text to: “The affected region is also home to several hydropower plants with water storage reservoirs, and…”

  • Line 67 - the "to" in "this was not the case to the URPB" probably should be a "for".

Done.

  • Line 83 - "Despite those progress" should be changed to "Despite that progress".

Done.

  • Lines 85 & 86 - "And this is evidenced" -- I would cross out the "And" and start the sentence with "This is evidenced."

Done.

  • Line 441 - I would change it to "These data are not easily measured" (sorry, it's a pet peeve of mine as my students would probably tell you)

Done. Your pet peeve is welcome.

  • Lines 506 & 507 - It should probably be "less data have been enveloped."

Done.

  • Line 521 - the "that" in the beginning of that line should be "than".

Done.

Reviewer 3 Report

1) The method used here could not give accurate solution.
2) The model is not a proper one to explain the mechanism proposed.
3) Similar work have been published by many different authors.

I highly recommend against publishing this work.

Author Response

We are grateful for the time dedicated to reading the manuscript. 

  1. The supplementary material provides essential information about the area and the summary statistics for calibration and validation periods of the SWAT modeling for the streamflow stations (Table S2), showing the accuracy of the proposed solution.
  2. The supplementary material explained the parameter assessment, and the SWAT model parameterization and sensitivity analysis, meteorological, and basin modeling were extensively explained in section 2 of the manuscript.
  3. Our manuscript aims to fill the gap of advanced modeling system usage in Brazil's water resources management. Currently, the governance and political management of water resources and their decisions depend on statistical methods in databases that are difficult to handle or access. In addition, the management of water resources at the regional level is heavily dependent on decisions regarding the generation of energy by hydroelectric plants, even though the economy is dependent on several other sectors. For instance, the study region has a relevant agricultural vocation that can impact exports to countries that depend on Brazilian agriculture, and the poor management of its water resources can also be extrapolated to different parts of the world. And we would also like to mention that, despite the existence of similar works, this is a recurring problem and that further research is still necessary. We have faced water crises in the last few years, mainly caused by bad water management. We expect this manuscript could be an init of using dynamic hydrologic model usage in the water resources management by the governmental agencies as a crisis avoided, mitigation, and resilience management tool.

Round 2

Reviewer 3 Report

Similar work have been published by many different authors.

I highly recommend against publishing this work.

Author Response

Despite the refusal to accept our manuscript, we are grateful for the time dedicated to reading our work.